# Conformity between Pacifier Design and Palate Shape in Preterm and Term Infants Considering Age-Specific Palate Size, Facial Profile and Lip Thickness

**DOI:** 10.3390/children9060773

**Published:** 2022-05-25

**Authors:** Gwendolin Sistenich, Claudius Middelberg, Thomas Stamm, Dieter Dirksen, Ariane Hohoff

**Affiliations:** 1Department of Orthodontics, University of Münster, Albert-Schweitzer-Campus 1, 48149 Münster, Germany; claudius.middelberg@ukmuenster.de (C.M.); hohoffa@uni-muenster.de (A.H.); 2Department of Prosthetic Dentistry and Biomaterials, University Hospital Münster, Albert-Schweitzer-Campus 1, 48149 Münster, Germany; dieter.dirksen@ukmuenster.de

**Keywords:** preterm infants, term infants, palates, growth, pacifier, non-nutritive sucking

## Abstract

This retrospective case-control study is the first to examine the spatial conformity between pacifiers and palates in 39 preterm infants (12 females, 27 males) and 34 term infants (19 females, 15 males), taking into account the facial-soft-tissue profile and thickness. The shape of 74 available pacifiers was spatially matched to the palate, and conformity was examined using width, height, and length measurements. In summary, the size concept of pacifiers is highly variable and does not follow a growth pattern, like infant palates do. Pacifiers are too undersized in width, length, and height to physiologically fit the palate structures from 0 to 14 months of age. There are two exceptions, but only for premature palates: the palatal depth index at 9–11 months of age, which has no clinical meaning, and the nipple length at <37 weeks of age, which bears a resemblance to the maternal nipple during non-nutritive sucking. It can be concluded that the age-size concept of the studied pacifiers does not correspond to any natural growth pattern. Physiologically aligned, pacifiers do not achieve the age-specific dimensions of the palate. The effects attributed to the products on oral health in term infants cannot be supposed.

## 1. Introduction

Two types of sucking are observed in the newborn, nutritive sucking and non-nutritive sucking (NNS). Since NNS does not involve swallowing, it is less complex in terms of coordination than nutritive sucking [1]. This more basic movement pattern develops very early, stabilizes after birth, and is subject to the risk of turning into a sucking habit. The group of sucking habits includes, among others, the prolonged use of the pacifier.

This habit can be responsible for the following changes in primary dentition: anterior open bite, posterior crossbite, distal primary canine relationship, decrease in upper intercanine width, increased overjet, lip incompetence, a narrow palate, and a decrease in oral muscular tonicity [2], especially when the protective effect of breastfeeding is reduced [3]. Malocclusions in the primary dentition are common and can impair chewing function and facial aesthetics, which later has a negative impact on the quality of life [4,5]. Breastfeeding duration of 3 months and pacifier use up to 48 months increased the prevalence of malocclusion by 5 and 15 times, respectively, with 42% higher prevalence in preterm infants than in term infants [6]. Studies on the hardness and surface of pacifiers have shown that surface texture and force resistance during sucking acts through the local oral sensory environment and alters the modulation of suction [7,8]. The design and properties of pacifiers are, therefore, critical factors that can alter the oral anatomy of premature and mature infants.

However, not only negative effects are attributed to the pacifier. There is evidence that NNS on a nonlactating nipple is effective in reducing pain-related reactivity during invasive examinations, such as a needle puncture [9,10]. These results are controversial, in that breastfeeding, breast milk, and glucose/sucrose administration were found to be more effective [11]. In addition, when using a pacifier, a shortened time to transition to full breastfeeding, to the time to discharge, and to oral feeding is observed [12,13]. For preterm infants, the pacifier has no effect on hospital stay, but their transition to full enteral or bottle feeds was easier [14]. The protective effect of pacifiers on sudden infant death syndrome (SIDS) is, also, discussed differently in the scientific literature [15]. If no evident correlations can be found on the basis of randomised controlled trials [16], more recent reviews point to a positive effect [17]. Even though there are other preventive measures against SIDS, it is understandable that parents continue to offer the pacifier because of its calming effect. Since SIDS has its highest incidence at 2–4 months of age and is already at zero after one year, it makes no sense to offer the pacifier beyond the age of two years [18].

The decision about the pacifier is a decision of the parents, who are exposed to the market recommendations and have free choice. Knowing that a pacifier can also have negative effects, one strives to choose the most physiological and healthy product for their child. Manufacturers advertise their products as “orthodontic”, “anatomical”, “jaw-fitting”, and many more, up to protective effects such as “prevention of tooth and jaw misalignment”. It would be desirable if the products would satisfy the natural reflex of NNS, without harmful influence and with support of a natural oral development. Unfortunately, it is, so far, unclear how these well-intentioned effects are achieved by the shape and size. There is, currently, a lack of information on what types of infant measurements have been used in the design of pacifiers, to meet the promised capabilities.

The aim of this study is, therefore, to investigate the conformity between pacifier design and palate shape, in preterm and term infants. To assess the pacifier shape in the physiological position to the palate, the age-specific palates, facial profiles, and lip thicknesses were considered. We hypothesize that there is an age-specific conformity of shape and size between commercially available pacifiers and infant’s palates, which supports manufacturers’ claims of protective effects on orofacial development.

## 2. Materials and Methods

The present study is a retrospective case-control study, based on scanned maxillary plaster casts from a clinical trial registered under ClinicalTrials.gov (NCT00408746). The plaster casts were made from upper jaw impressions taken from preterm and term infants, born at the tertiary-care University Hospital of Münster, Germany. Impressions were made within the first week after birth and, consecutively, at different examination times over the course of several quarters. Ethical approval was obtained from the local ethics committee prior to inclusion of the infants (13 January 1999) and for using retrospective anonymised data (4 September 2013).

Pacifiers from different manufacturers were chosen for conformity evaluation. Inclusion criteria were freely availability on the market, either via the Internet or in local stores, and the presence of certain oral health-related benefits (claims) that the manufacturers advertised with the product. In addition, clinical products (for hospital use only) for preterm infants were included, if the manufacturer provided them upon request.

### 2.1. Evaluation of Age-Related Palate Size and Shape

The data basis of the present study were the orthodontic plaster casts (class IV plaster, Silky Rock White, Whip Mix^®^, Louisville, KY, USA) of preterm and term infants from NCT00408746 (clinicaltrials.gov). The dental casts were digitised with the ATOS II^®^ system (GOM GmbH, Braunschweig, Germany) and stored in the standard tessellation language file format (STL), for further processing. Data processing and analysis as well as the theoretical background are, clearly, described in former publications [19,20]. Therefore, only a brief overview is given here.

The size and the shape of the palate varies considerably, with gestation and postnatal age [21]. The use of anatomical landmarks is too inaccurate, so consistent structures (features) were used for measurement. These are the crest of the alveolar ridge and the deepest point of the palatal vault. A prerequisite for reproducible automatic measurement is the identical alignment of the palates to a reference plane. It is, therefore, necessary to transform the object coordinate system into a reference coordinate system that provides identical measurement conditions for all consecutive plaster casts. We, therefore, used a coarse, landmark-dependent pre-transformation and a landmark-independent fine adjustment, to align the alveolar ridge, symmetrically, to the horizontal plane [19].

The reliability of the method has already been proven. The error ranges from 0.56% to 2.66% for feature-dependent, linear distances [19]. The measurements used in the present study comprised palatal width, palatal depth, and the distance in anteroposterior direction from the most anterior point of the alveolar process to the point of maximum palatal width as well as to the maximum palatal depth (Figure 1).

### 2.2. Orientation of Pacifiers into the Palatal Vault

Morphological references were sought to align the pacifiers, anatomically correctly, to the palatal vaults. It can be assumed that the alveolar ridge plane (ARP) in edentulous patients is parallel to the occlusal plane, which correlates highly to Camper’s line [22,23,24,25,26,27,28]. Camper’s line is defined as a line running from the tragus of the ear to the inferior border of the ala of the nose [29]. In young patients, the occlusal plane is found to be more parallel to Camper’s line, when the lower edge of the tragus is used as the posterior reference point [26].

Another reference used for pacifier orientation is the lip shield. It can be assumed that the shield lies in the plane Subnasale (Sn, the most superior and posterior point of the nasolabial curvature) to Pogonion molle (Po’, the most anterior point of the soft tissue chin). The angle between Camper’s line and Sn-Po’, therefore, indicates the correct angulation between ARP and the lip shield. To determine this angle, the profile images of the infants were graphically evaluated (Figure 2). With each examination, the infants’ photographs were taken with a Polaroid^®^ Spectra Macro 5 SLR 1200 instant film camera (Polaroid^®^, Minnetonka, MN, USA). Polaroid^®^ High Definition Grid Film, in the format 10.2 × 10.2 cm, was scanned with a Microtek Scanmaker^®^ (Microtek International^®^, Inc., Hsinchu, Taiwan), with a resolution of 500 dpi. Angular measurements were performed with Autodesk Inventor Professional 2021^®^ (Autodesk^®^, San Rafael, CA, USA).

A further reference is the soft tissue thickness below the contact surface of the lip shield. McKinnon et al. revealed that the soft tissues of the lips are significant correlated to most cranial dimensions [30]. This means that lip thickness increases with growth and, depending on age, the intraoral position of the pacifier’s artificial nipple changes. An upper lip thickness of 6.6 ± 0.3 mm for female newborns and 7.0 ± 0.4 mm for male newborns was reported. Within the first year, the lip thickness increases by 0.7 mm [30]. With these three references, the pacifiers were oriented in the median plane to the palates. In the vertical plane, the pacifiers were positioned until the first palatal contact.

### 2.3. Evaluation of Pacifier Size and Shape after Spatial Orientation to the Palate

The available pacifiers were scanned with the above-mentioned ATOS II^®^ scanning system. The length, width, and height of the artificial nipple was measured with Autodesk Inventor Professional 2021^®^. When the pacifier is placed into the mouth, the orientation and intraoral position of the nipple changes, depending on the position of the lip shield on the external soft tissues, which differs according to the curvature and thickness of the lips as well as the geometry of the lip shield. Therefore, the greatest vertical extent of the nipple into the palate (nipple depth) was measured after spatial orientation into the palatal vault. This, also, concerns the distances in anteroposterior direction from the lip shield to the point of maximum nipple width and to the maximum nipple depth (Figure 3).

### 2.4. Statistics

Descriptive statistics were performed using the software SPSS^®^ (IBM^®^ SPSS^®^ Statistics 27 for Mac, IBM Corp, Somers, NY, USA). To assess differences between palate and artificial nipple measurements, a one-way ANOVA was performed. Homogeneity of variances was tested using Levene’s test. In case of equal variances, Gabriel post-hoc test was chosen for multiple comparisons, otherwise the test according to Games-Howell was used. The Mann-Whitney U test was used to compare the two independent samples of preterm palate and preterm pacifier.

## 3. Results

### 3.1. Subjects

The study groups consist of 39 preterm infants (12 females and 27 males) and 34 term (19 females and 15 males) infants. Preterm and term infants are significantly different (*p* < 0.000), for nearly all baseline characteristics (Table 1). On the day of the first examination, the median age of the preterm infants was 36.0 weeks and that of the term infants was 40.43 weeks. Since pacifier sizes are based on age in months from birth, the ages of infants at each examination were calculated as follows: quarter 0 (0–2 months); quarter 1 (3–5 months); quarter 2 (6–8 months); quarter 3 (9–11 months); and quarter 4 (12–14 months). Preterm infants on the first day of examination were assigned to quarter −1.

The angle between Camper’s line and Sn-Po’ was almost identical, in preterm and term infants, at this time. From quarter 0, this angle increases to a median value of 80 degrees in preterm infants, whereas the angle in term infants remains almost unchanged, at a median of rounded 74 degrees in quarters 0–4.

### 3.2. Pacifiers

Seventy-four pacifiers, from a total of thirteen manufacturers, were included in this study, seventy of which are available for purchase and four of which are produced only for clinics (Appendix A). The size concept advertised by the manufacturers, which is based on the age of the child, has great similarities. For each month of a child’s life, excluding premature babies, between 17 and 37 different products for the same age are available for purchase (Appendix A). In total, there are three major size changes depending on age. The first change occurs after 2.6 ± 0.5 months, followed by a major product change at 6.1 ± 0.4 months. Another criterion of almost all manufacturers is that design change is followed by a longer period of stagnation in size. According to the manufacturer, the most common time period for which a pacifier can be used without changing to the next size is three months (19 products), followed by seven months (16 products). Interestingly, almost every time period is mentioned, except for 1, 2, 10, 17, 19, and 20 months. In one case, one size is even recommended for every age between 1 and 18 months.

As described above, the pacifiers were measured in analogy to the palates. This includes nipple width (*npw*, the widest part of the nipple), nipple depth (*npd*, the highest extension into the palatal vault), and the distance in anteroposterior (y-) direction from the lip shield (minus lip thickness) to the point of maximum nipple width (*y-npw*) as well as to the maximum nipple depth (*y-npd*). All measurements were made at spatial orientation to the palate and as a function of age-appropriate lip thickness. Due to the very different pacifier designs, many outliers and extreme values resulted. These data were trimmed, so that 62 pacifiers remained for further evaluation.

### 3.3. Palatal Depth

Projected onto a sagittal plane, the palate has the shape of a parabola, with the focal point below the deepest point. Anteriorly, the palate slopes towards the papilla and posteriorly the soft palate joins it, which also runs inferiorly. The hard-soft palate junction (HSPJ) is in close proximity to the deepest point of the palatal groove. Ultrasound images (Figure 4) show that the maternal nipple is in contact to the palate during non-nutritive suckings and is located at a distance of 8.1 ± 0.3 mm from the HSPJ [31,32], making the deepest point an anatomically important region.

The palatal depth (*pd)* is the distance in the vertical (z-) direction between ARP and the uppermost point of the palatal vault. This distance increases with age and is observed in both preterm and term infants. This increase is not observed in the design of the pacifiers. None of the pacifiers studied come close to achieving the measured palatal depth (Figure 5).

Due to the growth of the upper jaw forward and downward, the point of the deepest palate, indirectly, shifts to the back of the upper jaw. This increase is seen in both preterm and term infants. Even with this parameter, the basic design of the pacifier does not change. There is a slight increase in this distance over the quarters, but the values of the preterm and term palates are not reached, by far (Figure 5).

### 3.4. Palatal Width

The edentulous upper arch of a newborn infant has the shape of an ellipse, i.e., from anterior to posterior, it widens to a maximum transversal dimension (Figure 1) and narrows again. In normally growing children, the palatal width increases with age. This pattern is evident in both term and preterm infants (Figure 6). For pacifiers, there is an increase in nipple width over the quarters, but, similar to palatal depth, the infant’s measurements are not reached.

The widest part of the upper jaw, also, shifts indirectly backward as it grows. Figure 6, again, shows this growth pattern in preterm and term infants. This growth pattern is not evident in the design of the pacifiers. The widest part seems to remain at the same distance from the lip shield over all quarters.

### 3.5. Palatal Depth and Palatal Width Index

Palatal depth and palatal width indices are dimensionless parameters that represent the relationship between two distances in different planes. This ratio is insufficient for a description of growth [19], but it can characterize a basic form, regardless of whether the actual quantities are, also, correct.

Palatal depth index is the ratio between the depth of the palate (*pd*) and the distance to it (*y-pd*). Figure 7 confirms that the palatal depth index does not show a typical growth pattern across quarters. The values of the pacifiers are significantly smaller than those of the infants (*p* < 0.000), except for preterm infants in quarter 3 (equal variances assumed, *p* = 0.504). Seventeen pacifiers in quarter 3 (*av618*, *ba318*, *baa318*, *baas318*, *bf2*, *cpc616*, *de614*, *ga6+*, *gnr411*, *gns314*, *gr618*, *ma616*, *mp6-16*, *nic6+*, *nl518*, *nu618*, *nug618*) have a similar depth index as preterm infants, but without corresponding to the actual dimensions of *pd* and *y-pd*. None of the pacifiers mentioned are, specifically, designed for quarter 3.

Palatal width index is the ratio between the width of the palate (*pw*) and the distance to it (*pw*). Similar to the depth index, the palatal width index, also, shows no growth pattern over the quarters (Figure 7). The measurements of the pacifiers show significantly larger values (*p* < 0.05) in quarters 0–4, compared to palatal measurements. Only quarter −1 shows no difference in the width index (Mann-Whitney U test, *p* = 0.975). Pacifiers in this quarter are *bff*, *cus*, *jpp*, *nup00*, *nup01*, *ws*, and *wt* whereas only *jpp*, *wt*, and *bf* correspond to the preterm values.

### 3.6. Pacifier Length

As mentioned above, ultrasound images suggest that the maternal nipple extends to the deepest point of the palate, during non-nutritive sucking. It is only during breastfeeding that the nipple is pulled towards the HSPJ, by negative pressure. To simulate this position of the nipple during non-nutritive sucking, the pacifier should end at the deepest curvature of the palate. Figure 8 shows, once again, the distance from the most anterior to the deepest point of the palate, compared to the corrected pacifier length (angled, minus age-appropriate lip thickness). The measurements between infants and pacifiers differ significantly (*p* < 0.05) for quarters 1–4. Only preterm infants in quarter −1 (Mann-Whitney U test, *p* = 0.427) and quarter 0 (equal variances assumed, *p* = 0.874) show no differences between measurements. For quarter −1 these are: *bff*, *cus*, *jpp*, *nup00*, *nup01*, *ws*, and *wt*.

In quarter 0, 25 pacifiers have corresponding lengths to the depth of the palate, but only six products are specifically designed for quarter 0, i.e., the age group of 0–2 months. These are: *av02*, *ma02*, *mp02*, *nn02*, *nu02*, and *nug02*. The remaining 19 pacifiers have a fixed size for a longer period, of 0–3 and 0–6 months. These are: *av06*, *avs03*, *avu06*, *ba03*, *baa03*, *baas03*, *cpc06*, *cu07*, *de06*, *gno06*, *gnr03*, *gns03*, *gr06*, *ma06*, *mp06*, *nic06*, *nl06*, *nu06*, and *nug06*.

## 4. Discussion

In the present study, commercially marketed pacifiers were compared, in terms of their design, with the palatal shape of preterm and term infants. To the best of our knowledge, this is the first study to investigate the intraoral position of pacifiers, taking into account the child’s age-specific palate size, facial profile, and lip thickness.

Although more than 70 different products were included in this study, no claim to completeness of the current market coverage can be made. The focus was, primarily, on products with a statement about the natural similarity or fit to the child’s intraoral structures, which ultimately allows a metric comparison. A common feature of the manufacturers is that one or more product lines are offered for different age groups. The subdivision, for all products, refers to the age specification in months. Size recommendations by age are completely different between manufacturers, so parents can choose from 37 different products for a particular age (month) of a child. The further purchase decision is then subject to other parameters, such as the protective effect on the orofacial development. The protective effect is usually emphasized by the words “orthodontic”, “jaw-fitting”, “healthy or natural oral development”, or “anatomically correct”.

A prerequisite for an anatomically correct pacifier is an age-appropriate size, which corresponds to the natural growth process of the child’s palate. Looking at the size concepts of the manufacturers, no typical growth curve of pacifier sizes can be identified. Appendix A shows that the major product changes occur after 2.6 ± 0.5 months and 6.1 ± 0.4 months, followed by a longer period of stable dimensions. The second-most-common latency between two sizes is seven months. This stagnation in orofacial growth is not observed in either preterm or term infants [33,34].

A study on 17 Japanese term infants [35] observed the same growth values and pattern of palatal width and depth as in our study. Growth of the palates did not occur, consistently, in all spatial directions and showed an insignificant increase in palatal depth, within the first year. A flattening of the growth curve for palatal depth was, also, confirmed in our study, as well as in a study by Zen et al. [36]. Therefore, a physiological growth effect in favor of other structures can be assumed.

Another prerequisite for an anatomically correct pacifier should be a certain dimensional similarity to the child’s palate. Objectively measurable parameters, which allow conclusions to be drawn about natural growth, are the dimensions in the three spatial directions, whereby reproducible measuring points should be used [19,20]. The dimensional similarity to the child’s palate must be present in the correct position. This is determined by the pacifier’s lip shield on the child’s lip profile and the lip thickness, which limits the intraoral position of the artificial nipple in the anteroposterior direction. The lip profile and lip thickness have not been considered in studies to date. One exception is a finite element analysis (FEA) study [37], in which a lip thickness of 6.6 mm was considered using a representative edentulous palate of unknown age. A recent FEA study considered a lip thickness of 5 mm at an angle of 5–10 degrees [38]. Our own results indicate an angle of 16.29 degrees.

Ultrasound measurements, about tongue movements and breastfeeding, show that the maternal nipple is located in the highest curvature of the palate during non-nutritive sucking and just before the onset of nutritive sucking [31,32,39]. The present study shows that this depth is not reached by pacifiers and, therefore, an anatomically correct position in vertical direction cannot be assumed. This also concerns the horizontal position of the highest extension of the nipple, which is much too far anterior. No age-specific size concept comes close to the values of the palates.

Similar relationships to the depth of the palate exist in the transverse width of the pacifiers. The artificial nipples are, usually, approximately half the width of the widest part of the palates. As a result, the lateral tektal bulges and walls [40], in particular, are not supported. FEM studies show that in small nipple shapes, the load is concentrated on the central zone of the palate, which cannot guarantee the preservation of the transverse diameter of the premaxilla [38,41].

If the position of the largest width in the anteroposterior direction is considered, the pacifiers examined are, also, much too short in this respect. This applies to all the products investigated, including the so-called orthodontic nipples, which promise better support. Here, too, there is no recognizable size concept that corresponds to a natural growth pattern.

Even if the manufacturers’ size concepts do not represent a growth pattern and are far too undersized, the basic design may be physiological. To assess this, the depth and width index was calculated. The relation of the vertical extent of the nipple and the position in the anteroposterior direction to the palate (palatal depth index) is fundamentally different from the measurements of term infants. Again, the basic design is too small. An exception is age quarter 3, in which there is no significant difference to the palates of the preterm infants. However, it must be considered that this only concerns preterm infants, and none of the pacifiers are designed for preterm infants or for this age. Similar relationships affect the palatal width index. Only the preterm pacifiers do not differ significantly from the width index of the preterm palates. This indicates a favorable width relation, but without representing the real sizes.

The undersizing of pacifiers is, also, significant in terms of different populations. Zen et al. investigated the palatal width and depth on 80 Brazilian term newborns at birth and six months of age [36]. Although the measurement points were not identical to the present study, they were able to determine significantly larger median values for the width and depth of the palate. While the increase in palatal depth from birth to six months of age was nearly identical to our results (difference = 0.2 mm), there was a significantly larger increase in palatal width (difference = 6.8 mm) in the Brazilian population. The results indicate that the demands on a pacifier design can be very high, when manufacturers advertise their products as “healthy” or “anatomically correct”.

Other important parameters are the length of a pacifier and how far the artificial nipple reaches into the oral cavity. Since the pacifiers studied are designed only for the need of non-nutritive sucking, the length should end far before the HSPJ. During non-nutritive sucking, the maternal nipple is located at the deepest curve of the hard palate and at a distance of 8.1 ± 0.3 mm from the HSPJ [31,32]. An anatomically correct length of an artificial nipple would, therefore, be the distance from the most anterior to the deepest point of the palate. Overall, again, all products are far too undersized for term infants in each age quarter.

Only for preterm infants in quarter −1 (<37 weeks of gestation) and 0 are comparable lengths observed. Regarding quarter 0, it should be noted that the similarity in length between pacifiers and preterm infants’ palates has no clinical relevance, as pacifiers are designed exclusively for term infants.

However, the pacifiers specifically designed for preterm infants should be evaluated differently, as they are used to reinforce non-nutritive sucking behaviour and to enhance the sucking experience, and, thus, do not meet the goals of pacifiers for term infants. Although there is no statistical difference in quarter −1, the lengths of the pacifiers compared to the palates vary considerably (Figure 9).

The need for an age-appropriate pacifier dimension is further emphasized by the relationship between palate shape and sudden unexpected death in infancy (SUID). SUID includes deaths from sudden infant death syndrome (SIDS), which are deaths that remain unexplained after complete post-mortem investigations [42,43]. A recent computed tomography and autopsy study of children, who died of SUID at an average age of five months, showed that the SUID group had significantly narrower palates than the control group [43]. A pacifier that is too small could be an additional risk factor for palatal predisposition, especially if there are other habits, usually unnoticed, that could have a possible influence on the orofacial structures.

Environmental factors and genetic causes are discussed in relation to habits [36]. For example, the weaned pacifier may be replaced by the thumb and exacerbate a narrow palate. The transitions to habitual oral respiration are smooth, and the cause and effect of nasal obstruction is controversial in this context. The same applies to neuromuscular deficiency [44], head posture [45], and rheumatoid disease [46]. A mutual influence can be assumed, so that the use of a “wrong” pacifier may overload a child’s ability to compensate.

This study has limitations that must be considered when interpreting the results. The physiological positioning of the pacifiers, taking into account the facial profiles and soft tissue thickness, was static based on the STL data used. Due to the flexibility of the pacifier materials and soft tissue, positional changes may occur in the oral cavity that cannot be accounted for by the present method.

## 5. Conclusions

Based on the results of this study, the hypothesis that there is an age-specific conformity of shape and size between commercial pacifiers and the palates of infants must be rejected. The following conclusions could be drawn:The existing size concept of the investigated pacifiers does not indicate a growth pattern in any of their dimensions, as observed in normally growing preterm and term palates and, therefore, cannot be considered physiological.Anatomically correctly aligned to the palate, the investigated pacifiers appear undersized in their basic dimensions. Even if elastic deformation of the nipple and lips during sucking would be taken into account, there is no natural fit to the palate.The design relations of height, width, and length show hardly any similarities to the palatal parameters. Based on the results, the oral health effects attributed to the products for term infants cannot be supposed.Existing preterm pacifiers have great potential. A separate product line with special characteristics for premature infants, even beyond the 40th week, seems to be reasonable.

## Figures and Tables

**Figure 1 children-09-00773-f001:**
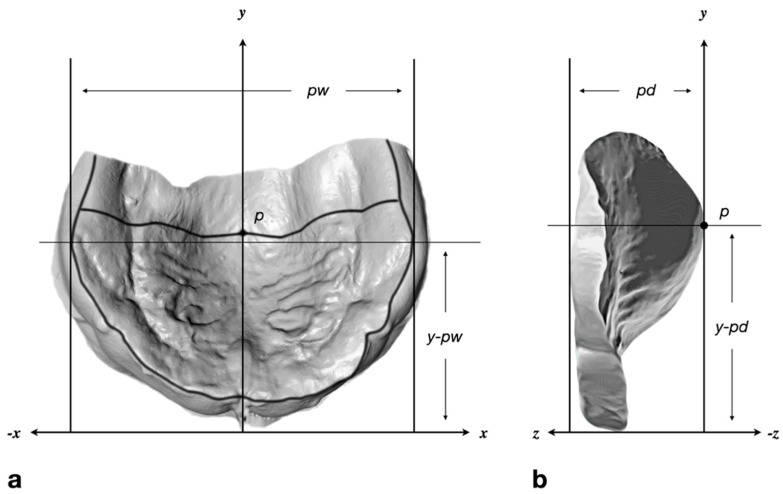
Measurements obtained from digitized plaster casts. (**a**) Inferosuperior and (**b**) lateromedial view of a newborn’s palate. All areas of the scanned plaster cast that were not covered by impression material were digitally removed. The segmented casts were oriented in a 3D coordinate system, according to the raphe palatina mediana, and by symmetrical alignment of the alveolar ridge towards a horizontal reference plane. The most inferior points (z-direction) of the alveolar bone constitute the alveolar ridge (black line). Point *p* is the deepest point of the palatal vault. Palatal width (*pw*) is the longest distance in the x-direction and perpendicular to the y-axis between two surface points on the right and left side of the alveolar ridge. Palatal depth (*pd*) is the distance in the z-direction, between the most inferior and most superior point of the palatal shape. Further measurements are the distance in the anteroposterior (y-) direction from the most anterior point of the alveolar process to the point of maximum palatal width (*y-pw*) and to the maximum palatal depth (*y-pd*).

**Figure 2 children-09-00773-f002:**
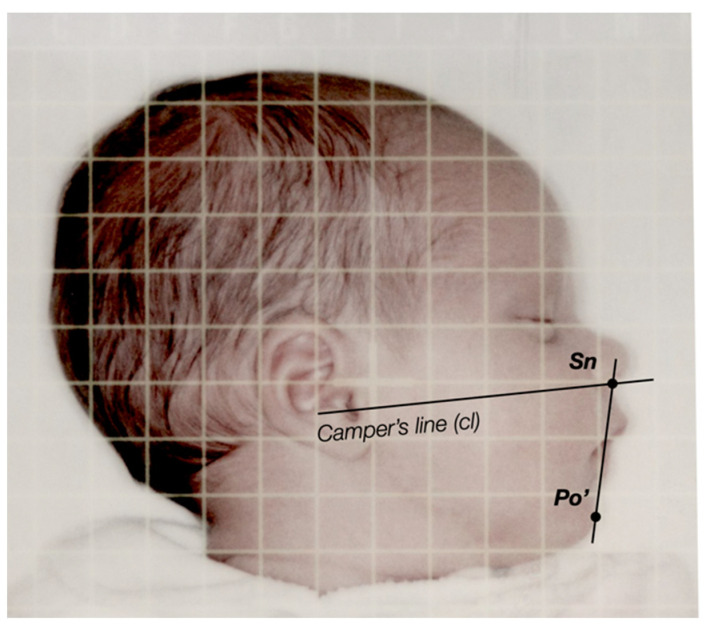
Examination of the infants’ profile, according to Camper’s line (cl). Camper’s line in infants is determined by a line running from the lower edge of the tragus of the ear to the inferior border of the ala of the nose. The lip shield covers the upper lip and lower lip of the child. These have different thicknesses depending on age, but orientation is based on the anthropometric landmarks Subnasale (Sn) and Pogonion molle (Po’).

**Figure 3 children-09-00773-f003:**
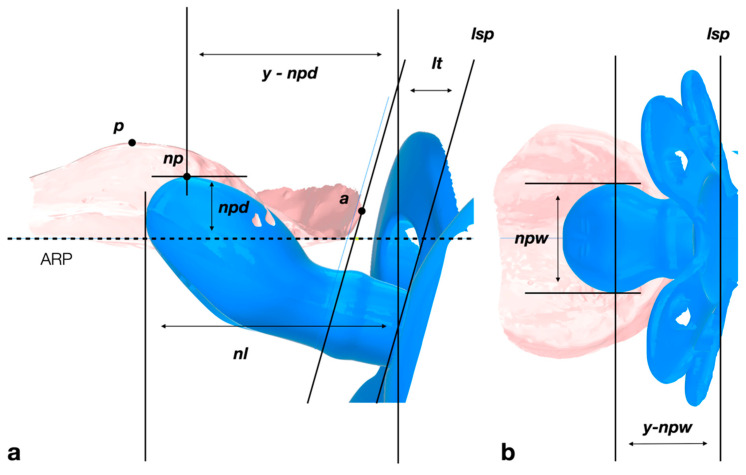
Measurements according to spatial orientation of the pacifier on the infant’s palate. (**a**) Lateromedial view on a median section through palate and pacifier. The pacifier is angulated according to the angle between the alveolar ridge plane (ARP, a parallel to Camper’s line) and the lip shield plane (*lsp*). The horizontal orientation is determined by the lip thickness (*lt*), a distance between the lip shield plane (*lsp*) and the most anterior point of the palate (*a*). The vertical position is determined by the first contact to the palatal shape. In this orientation, the nipple’s palatal depth (*npd*) is the distance between ARP and the most superior point of the nipple. The length *y-npd* is the distance between point *a* and the highest point of the nipple (*np*) in the anteroposterior (y-) direction, parallel to ARP. The nipple length (*nl*) is the distance between the lip shield plane and the most posterior border of the nipple, parallel to ARP. (**b**) Inferosuperior view on the palate is according to the left figure. The pacifier is aligned towards the median plane of the palate. The nipple’s palatal width (*npw*) is the largest width of the pacifier, and *y-npw* is the distance from *npw* to *lsp*.

**Figure 4 children-09-00773-f004:**
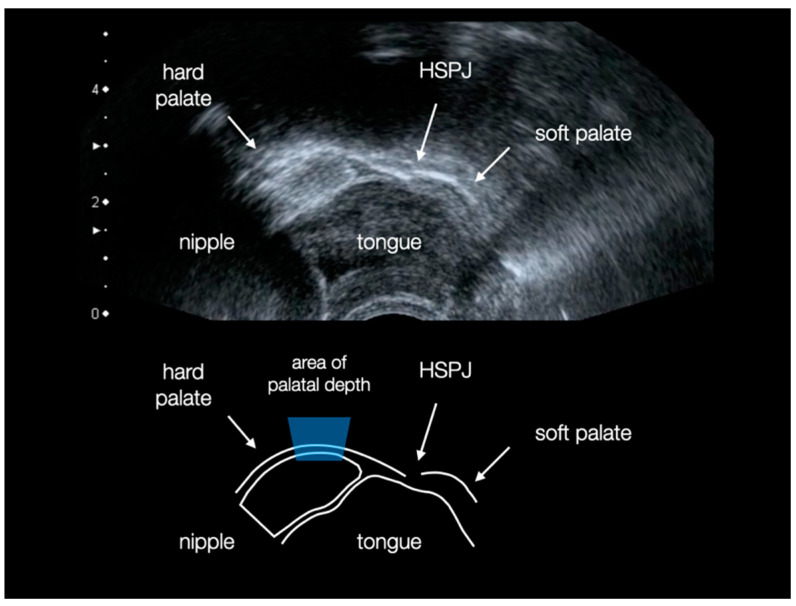
Midline submental ultrasound image of an infant’s oral cavity during non-nutritive sucking. The tongue is in a superior position and the maternal nipple rests at the deepest curvature of the hard palate (area of palatal depth). When nutritive sucks are performed, the nipple increases in volume and reaches up to 2.7 ± 1.0 mm to the hard-soft palate junction (HSPJ).

**Figure 5 children-09-00773-f005:**
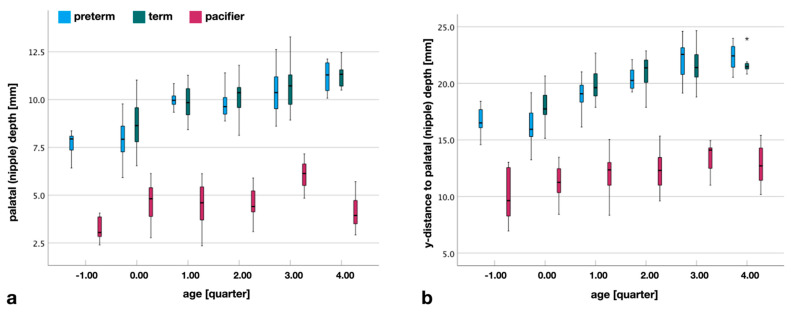
(**a**) Palatal depth (pd) and artificial nipple depth (*npd*). Distance in vertical (z-) direction between ARP and the deepest point of the palate (preterm, term) as well as the highest point of the artificial nipple. Compared to the infants’ palates, the pacifiers do not show any increase in size, in terms of a growth effect. (**b**) Location of the deepest point of the palate and the highest point of the artificial nipple. Distance in the horizontal (y-) direction from the anteriormost to the deepest part of the palate (preterm, term) and from the lip shield plane to the highest point of the artificial nipple minus lip thickness (pacifier). (*) extreme outlier.

**Figure 6 children-09-00773-f006:**
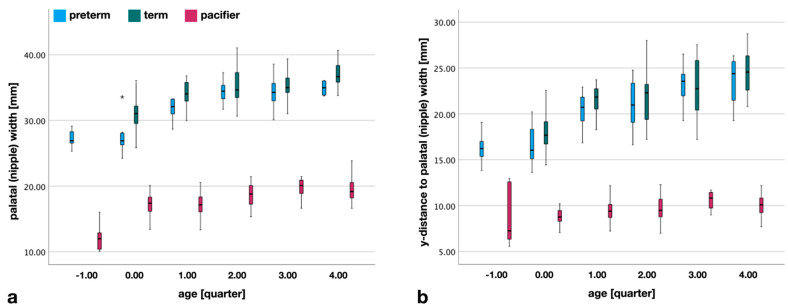
(**a**) Palatal width (*pw*) and artificial nipple width (*npw*). The artificial nipple widths show no similarities to the palates. Despite a small increase over time, a growth pattern is not visible. Preterm pacifiers are particularly narrow in dimension. (**b**) Location of the widest part of the infant’s palate and artificial nipples. Distance in the horizontal (y-) direction from the anteriormost to the widest part of the palate (preterm, term) and from the lip shield plane to the widest part of the artificial nipple minus lip thickness (pacifier). Although the width changes somewhat, the distance to the lip shield does not seem to have been considered in the design of the pacifiers. (*) extreme outlier.

**Figure 7 children-09-00773-f007:**
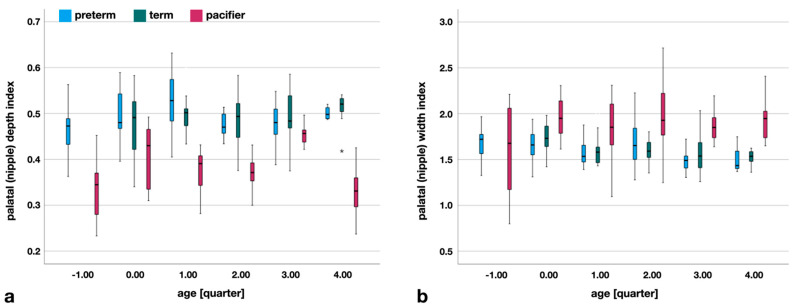
(**a**) Palatal depth index, a dimensionless parameter of the ratio of *pd* to *y-pd*, with no meaning for growth. Pacifier scores are significantly smaller than infant scores, except in quarter 3, and are only related to preterm infants. (**b**) Palatal width index as a ratio of *pw* to *y-pw*. Here, the pacifier values are significantly larger, except in quarter −1. (*) extreme outlier.

**Figure 8 children-09-00773-f008:**
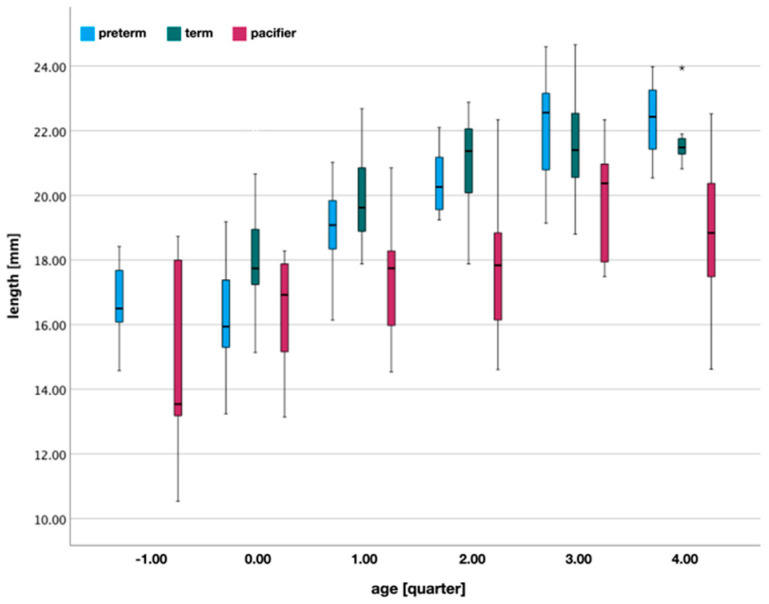
Pacifier (intraoral) length, in relation to the anteroposterior distance to the deepest point of the palatal vault. The infant palate values correspond to the position of the maternal nipple during NNS. The lengths of pacifiers are, significantly, too short compared to the palates of term infants. This does not apply to the palates of preterm infants in quarter −1 and 0. (*) extreme outlier.

**Figure 9 children-09-00773-f009:**
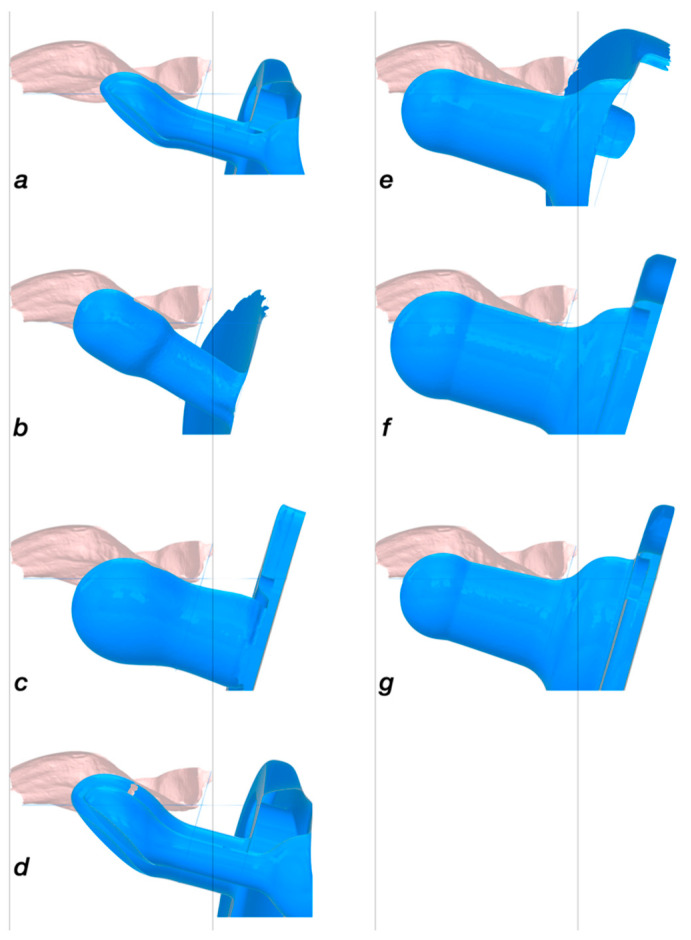
Preterm pacifiers placed into a palate of a female preterm infant of 35 weeks of gestational age. The length of the artificial nipples and the basic design vary considerably. For example, the shortest artificial nipple a is 4.04 mm shorter than the smallest and 7.88 mm shorter than the largest preterm measurement in this group. (**a**) = nup01; (**b**) = wt; (**c**) = bff; (**d**) = nup00; (**e**) = jpp; (**f**) = cus; (**g**) = ws.

**Table 1 children-09-00773-t001:** Baseline characteristics of preterm and term infants on the first day of examination. Mean and 95% confidence intervals of gestational age, weight, head circumference (hc), and body length at birth. Median of the profile angle between Camper’s line (Cl) and a line between the landmarks Subnasale (Sn) and Pogonion molle (Po’). Except for the profile angle (Cl-(Sn-Po’)), all values are significant differently (*p* < 0.000) between both groups.

Infants	Female	Male	Age [Weeks] Mean (95% CI)	Weight [kg] Mean (95% CI)	hc ^1^ [cm] Mean (95% CI)	Length [cm] Mean (95% CI)	Cl-(Sn-Po’) ^2^ Median
preterm	12	27	30.33 (29.37–31.29)	1.44 (1.26–1.61)	27.29 (26.24–28.35)	38.63 (36.96–40.29)	73.85
term	19	15	39.32 (38.86–39.78)	3.47 (3.29–3.65)	34.90 (34.46–35.33)	50.76 (49.82–51.71)	73.71

^1^ head circumference; ^2^ [degrees].

## Data Availability

Not applicable.

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
