# Peer review of "Conformity between Pacifier Design and Palate Shape in Preterm and Term Infants Considering Age-Specific Palate Size, Facial Profile and Lip Thickness"

_children, 2022, doi:10.3390/children9060773_

Round 1

Reviewer 1 Report

The manuscript titled “Conformity between pacifier design and palate shape in preterm and term infants considering age-specific palate size, facial profile and lip thickness” compares the various commercially available pacifier designs and their influence on the palatal shape and Orofacial morphology. The manuscript is well written and scientifically sound. I do have few doubts and suggestions which could improve the quality of the existing good work and will be interesting for the readers. 
  1. Will there be any changes in the data with gender? 
  2. Why sample size of preterm and tern different for males and females?
  3. Few more references and details of the statement of problem using different pacifiers can be added in the introduction. 
  4. Introduction: State the hypothesis of the study in the last paragraph after the aims.
  5. Was the palatal depth and width consistent for each digital cast models? 
  6. Does the material selected for pacifier( either soft or hard) will influence the shape of the palate? 
  7. Include recent reference studies in the discussion to compare the data obtained with other studies. Justify how this study is different and data is sustainable. 

Author Response

Dear Reviewers,
Thank you for taking the time to review our manuscript. We are grateful for the overall positive feedback and cherish your constructive suggestions and the valuable input, that will help improve our manuscript. We hope that your concerns and suggestions were addressed appropriately. In the following we give a point-by-point reply to your comments.

Reviewer's note 1: Will there be any changes in the data with gender?
Response: Dear reviewer, although female and male infants differ significantly in their baseline characteristics, there are no significant differences in palatal measurements. The statements in comparison to the pacifiers therefore do not change if the groups had been analyzed separately.
Reviewer's note 2: Why sample size of preterm and tern different for males and females?
Response: Dear reviewer, recruiting infants, or convincing parents to participate in a study unrelated to their child's current health status, was one of the most difficult tasks. Among premature infants, twin and triplet births were more common, therefore the sample size is different.
Reviewer's note 3: Few more references and details of the statement of problem using different pacifiers can be added in the introduction.
Response: Dear reviewer, we added the following to the introduction:
... , especially when the protective effect of breastfeeding is reduced (Peres 2015). Malocclusions in the primary dentition are common and can impair chewing function and facial aesthetics, which later has a negative impact on the quality of life (Peres 2007, Kramer 2013). Breastfeeding duration of 3 months and pacifier use up to 48 months increased the prevalence of malocclusion by 5 and 15 times, respectively, with 42% higher prevalence in preterm than in term infants (da Rosa 2020). Studies on the hardness and surface of pacifiers
have shown that surface texture and force resistance during sucking acts through the local oral sensory environment and alters the modulation of suction (Zimmermann 2008, Oder 2013). Design and properties of pacifiers are therefore critical factors that can alter the oral anatomy of premature and mature infants.
-da Rosa DP, Bonow MLM, Goettems ML, Demarco FF, Santos IS, Matijasevich A, et al. The influence of breastfeeding and pacifier use on the association between preterm birth and 2 primary-dentition malocclusion: A population-based birth cohort study. Am J Orthod Dentofacial Orthop [Internet]. 2020;157:754–63.
-Peres KG, Cascaes AM, Nascimento GG, Victora CG. Effect of breastfeeding on malocclusions: a systematic review and meta-analysis. Acta Paediatr [Internet]. 2015;104:54–61.-Peres KG, Barros AJD, Peres MA, Victoria CG. Effects of breastfeeding and sucking habits on malocclusion in a birth cohort study. Rev Saude Publica. 2007;41:343–50. -Kramer PF, Feldens CA, Helena Ferreira S, Bervian J, Rodrigues PH, Peres MA. Exploring the impact of oral diseases and disorders on quality of life of preschool children. Community Dent
Oral Epidemiol [Internet]. 2013;41:327–35. -Zimmerman E, Barlow SM. Pacifier stiffness alters the dynamics of the suck central pattern generator. J Neonatal Nurs [Internet]. 2008;14:79–86.
-Oder AL, Stalling DL, Barlow SM. Short-Term Effects of Pacifier Texture on NNS in Neurotypical Infants. Int J Pediatr. 2013;2013:1–8.
Reviewer's note 4: Introduction: State the hypothesis of the study in the last paragraph after the aims.
Response: Dear reviewer, we added the following after the aim of our study:
We hypothesize that there is an age-specific conformity of shape and size between commercially available pacifiers and infant’s palates that supports manufacturers' claims of protective effects on orofacial development.
We also added the following to the conclusion section: Based on the results of this study, the hypothesis that there is an age-specific conformity of shape and size between commercial pacifiers and the palate of infants must be rejected.
Reviewer's note 5: Was the palatal depth and width consistent for each digital cast models?
Response: Dear reviewer, the measurements were not consistent for each digital cast as seen in the boxplot figures.
Reviewer's note 6: Does the material selected for pacifier (either soft or hard) will influence the shape of the palate?
Response: Dear reviewer, studies confirm that the material and also the surface texture have an effect on the modulation of sucking and thus on the oral anatomy. We have included this in the introduction, as written under reviewers’s note 3. FEA studies confirm these effects (please see discussion). Unfortunately, we worked with STL meshes, which do not allow any conclusions to be drawn about the force acting on the palate and the associated deformations.
Reviewer's note 7: Include recent reference studies in the discussion to compare the data obtained with other studies. Justify how this study is different and data is sustainable.
Response: Dear reviewer, in agreement with other authors, only few current studies on palatal growth and pacifiers can be found in the literature. Results of the following recent studies are included in the discussion:
A study on 17 Japanese term infants (Kihara 2017) observed the same growth values and pattern of palatal width and depth as in our study. Growth of the palates did not occur consistently in all spatial directions and showed a insignificant increase in palatal depth within the first year. A flattening of the growth curve for palatal depth was also confirmed in our study, as well as in a study by Zen et al. 2020. Therefore, a physiological growth effect in favor of other structures can be assumed. 3 The undersizing of pacifiers is also significant in terms of different populations. Zen and coworkers investigated the palatal width and depth on 80 Brazilian term newborns at birth and six month of age (Zen 2020). Although the measurement points were not identical to the present study, they were able to determine significantly larger median values for the width and depth of the palate. While the increase in palatal depth from birth to 6 months of age was nearly identical to our results (difference = 0.2 mm), there was a significantly larger increase in palatal width (difference = 6.8 mm) in the Brazilian population. The results indicate that the demands on a pacifier design can be very high when manufacturers advertise their products as "healthy" or "anatomically correct."
After line 338:
The need for an age-appropriate pacifier dimension is further emphasized by the relationship between palate shape and sudden unexpected death in infancy (SUID). SUID includes deaths from sudden infant death syndrome (SIDS), which are deaths that remain unexplained after complete post-mortem investigations (Rambaud 2012, Ducloyer 2022). A recent computed tomography and autopsy study of children who died of SUID at an average age of five months showed that the SUID group had a significantly narrower palates than the control
group (Ducloyer 2022). A pacifier that is too small can therefore be an additional risk factor for a predisposed palate.
-Kihara T, Kaihara Y, Iwamae S, Niizato N, Gion S, Taji T, et al. Three-dimensional longitudinal changes of maxilla and mandible morphology during the predental period. Eur J Paediatr Dent. 2017;18:139–44.
-Zen I, Soares M, Pinto LMCP, Ferelle A, Pessan JP, Dezan-Garbelini CC. Maxillary arch dimensions in the first 6 months of life and their relationship with pacifier use. Eur Arch Paediatr Dent [Internet]. Springer Berlin Heidelberg; 2020;21:313–9.
-Rambaud C, Guilleminault C. Death, nasomaxillary complex, and sleep in young children. Eur J Pediatr [Internet]. 2012;171:1349–58.
-Ducloyer M, Wargny M, Medo C, Gourraud P-A, Clement R, Levieux K, et al. The Ogival Palate: A New Risk Marker of Sudden Unexpected Death in Infancy? Front Pediatr [Internet]. 2022;10.

Reviewer 2 Report

I would like to congratulate authors for preparing this interesting paper titled "Conformity between pacifier design and palate shape in preterm and term infants considering age-specific palate size, facial profile and lip thickness".

The article deals with important topic in pedriatic dentistry concerning the influence of pacifier design and development of oral cavity and face.

Manuscript has been prepared properly, according to all science's papers requirements.

Well-described methodology of the research is followed by the adequate discussion over the results of the analyses.

 But some improvements are suggested. I provided more details below.

  • Abstract of the manuscript should be supplemented with the brief information concerning the novelty of the studies.
  • The results are adequately described. If you have same data on figures and in tables than omit one of them.
  • Discussion - this section  should be supplemented with brief information on the known other habbits and reasons that provides to malformations and compare it with obtained results.

Author Response

Dear Reviewers,
Thank you for taking the time to review our manuscript. We are grateful for the overall positive feedback and cherish your constructive suggestions and the valuable input, that will help improve our manuscript. We hope that your concerns and suggestions were addressed appropriately. In the following we give a point-by-point reply to your comments. 

Reviewer's note 1: Abstract of the manuscript should be supplemented with the brief information concerning the novelty of the studies.
Response: Dear reviewer, we modified the following sentences to meet the abstract criteria of 200 words or less.
This retrospective case-control study is the first to examine the spatial conformity between pacifiers and palates in 39 preterm (12 females, 27 males) and 34 term infants (19 females, 15 males), taking into account the facial soft tissue profile and thickness. The shape of 74 available pacifiers was spatially matched to the palate and conformity was examined using width, height, and length measurements. In summary, the size concept of pacifiers is highly
variable and does not follow a growth pattern like infant palates. Pacifiers are too undersized in width, length and height to physiologically fit the palate structures from 0 to 14 months of age. There are two exceptions, but only for premature palates: the palatal depth index at 9-11 months of age, which
has no clinical meaning, and the nipple length at < 37 weeks of age, which bears a resemblance to the maternal nipple during nonnutritive sucking. It can be concluded that the age-size concept of the studied pacifiers does not correspond to any natural growth pattern. Physiologically aligned, pacifiers
4 do not achieve the age-specific dimensions of the palate. The effects attributed to the products on oral health in term infants cannot be supposed.
Reviewer's note 2: The results are adequately described. If you have same data on figures and in tables than omit one of them.
Response: Dear reviewer, we have not presented data multiples times in figures and tables. Reviewer's note 3: Discussion - this section should be supplemented with brief information on the known other habbits and reasons that provides to malformations and compare it with obtained
results.
Response: Dear reviewer, your recommendation matches the recommendation of reviewer 1, so we have added the following text at the end of the discussion.
Environmental factors and genetic causes are discussed in relation to habits (Zen 2020). For example, the weaned pacifier may be replaced by the thumb and exacerbate a narrow palate. For example, the weaned pacifier may be replaced by the thumb and exacerbate a low open bite. The transitions to habitual oral respiration are smooth and the cause and effect of nasal obstruction is controversial in this context. The same applies to neuromuscular deficiency (Farronato 2013), head posture (Rijpstra 2016) and rheumatoid disease
(Sidebottom 2013). A mutual influence can be assumed, so that the use of a "wrong" pacifier may overloads a child's ability to compensate.
Farronato G, Giannini L, Galbiati G, Stabilini SA, Maspero C. Orthodontic-surgical treatment: neuromuscular evaluation in open and deep skeletal bite patients. Prog Orthod [Internet]. 2013;14:41. Rijpstra C, Lisson JA. Ätiologie des frontalen offenen Bisses: Ein Review. J Orofac Orthop. 2016;77:281–6.
Sidebottom AJ, Salha R. Management of the temporomandibular joint in rheumatoid disorders. Br J Oral Maxillofac Surg [Internet]. 2013;51:191–8.

Reviewer 3 Report

Dear Authors 

Paper is well designed and written well. It's a good study to raise a point pacifier design. 

It will be good if the authors add some latest papers in the introduction related to the core of the topic. 

Are there any limitations faced during this work?

Author Response

Dear Reviewers,
Thank you for taking the time to review our manuscript. We are grateful for the overall positive feedback and cherish your constructive suggestions and the valuable input, that will help improve our manuscript. We hope that your concerns and suggestions were addressed appropriately. In the following we give a point-by-point reply to your comments.

Reviewer's note 1: It will be good if the authors add some latest papers in the introduction related to the core of the topic.
Response: Dear reviewer, your recommendation matches the recommendation of reviewer 1. In agreement with other authors, only a few recent studies on the topic of palate growth and pacifiers can be found in the literature and these mainly refer to mature born infants. In addition to the recent FEA studies, we have included some recent clinical studies in the introduction and discussion. Please see first reviewer’s not 3 and 7. Reviewer's note 2: Are there any limitations faced during this work?
Response: Dear reviewer, yes, this study has limitations that should be considered when interpreting the results. We have added the following text at the end of the discussion.
This study has limitations that must be considered when interpreting the results. The physiological positioning of the pacifiers, taking into account the facial profiles and soft tissue thickness, was static based on the STL data used. Due to the flexibility of the pacifier materials 5 and soft tissue, positional changes may occur in the oral cavity that cannot be accounted for by the present method.
We appreciate the time and effort that you and the other reviewer dedicated. We tried our best to meet all the suggestions by the referees and improve the manuscript and hope that the correction will meet with approval. You have suggested that our manuscript undergo extensive English revisions.
Therefore, we have had our manuscript reviewed and corrected by an English-speaking colleague.
All co-authors approved the final version of the revision.
Thank you very much.
Yours sincerely,
Gwendolin Sistenich